# Spatial dilemmas of diffusible public goods

**Benjamin Allen[1,2]\*, Jeff Gore[3], Martin A Nowak[2,4,5]**

[1]Department of Mathematics, Emmanuel College, Boston, United States;
[2]Program for Evolutionary Dynamics, Harvard University, Cambridge, United States;
[3]Department of Physics, Massachusetts Institute of Technology, Cambridge, United
States; [4]Department of Mathematics, Harvard University, Cambridge, United States;
[5]Department of Organismic and Evolutionary Biology, Harvard University, Cambridge,
United States

**Abstract** The emergence of cooperation is a central question in evolutionary biology.
Microorganisms often cooperate by producing a chemical resource (a public good) that benefits
other cells. The sharing of public goods depends on their diffusion through space. Previous theory
suggests that spatial structure can promote evolution of cooperation, but the diffusion of public
goods introduces new phenomena that must be modeled explicitly. We develop an approach where
colony geometry and public good diffusion are described by graphs. We find that the success of
cooperation depends on a simple relation between the benefits and costs of the public good, the
amount retained by a producer, and the average amount retained by each of the producer's
neighbors. These quantities are derived as analytic functions of the graph topology and diffusion
rate. In general, cooperation is favored for small diffusion rates, low colony dimensionality, and
small rates of decay of the public good.

## Introduction

Public goods dilemmas are frequently observed in microbes. For example, the budding yeast
*Saccharomyces cerevisiae* cooperates by producing the enzyme invertase, which hydrolyzes sucrose
into monosaccharides, when yeast colonies are grown in glucose-limited media (*Greig and Travisano,
2004*; *Gore et al., 2009*). Other examples include the production of chemical agents that scavenge
iron (*Griffin et al., 2004*; *Buckling et al., 2007*; *Cordero et al., 2012*; *Julou et al., 2013*), enable
biofilm formation (*Rainey and Rainey, 2003*), eliminate competition (*Le Gac and Doebeli, 2010*),
induce antibiotic resistance (*Chuang et al., 2009*; *Lee et al., 2010*), or facilitate infection of a host
(*Raymond et al., 2012*).

In many cases, the benefits of public goods go primarily to cells other than the producer. For example,
in a *S. cerevisiae* population subject to continuous mixing, only ~1% of monosaccharides are imported
into the cell that hydrolyzes them, with the remainder diffusing away (*Gore et al., 2009*). Furthermore,
production of public goods typically involves a metabolic cost, which may exceed the direct benefit to
the producer. In this case, absent some mechanism to support cooperation (*Nowak, 2006*), public
goods production is expected to disappear under competition from cheaters, resulting in the tragedy
of the commons (*Hardin, 1968*).

There is growing evidence from experiments (*Griffin et al., 2004*; *Kümmerli et al., 2009*; *Julou
et al., 2013*; *Momeni et al., 2013*) and simulations (*Allison, 2005*; *Misevic et al., 2012*) that spatial
or group clustering can support cooperation in microbial public goods dilemmas, although this effect
depends on the nature of competition for space and resources (*Griffin et al., 2004*; *Buckling et al.,
2007*). These findings agree with insights from mathematical models (*Nowak and May, 1992*; *Durrett
and Levin, 1994*; *Santos and Pacheco, 2005*; *Ohtsuki et al., 2006*; *Szabó and Fáth, 2007*; *Taylor et al.,
2007*; *Perc and Szolnoki, 2008*; *Fletcher and Doebeli, 2009*; *Korolev and Nelson, 2011*) suggesting

**\*For correspondence:**
benjcallen@gmail.com

**Competing interests:** The
authors declare that no
competing interests exist.

**Reviewing editor**: Carl T
Bergstrom, University of
Washington, United States

**eLife digest** The natural world is often thought of as a cruel place, with most living things ruthlessly competing for space or resources as they struggle to survive. However, from two chimps picking the fleas off each other to thousands of worker ants toiling for the good of the colony, cooperation is fairly widespread in nature. Surprisingly, even single-celled microbes cooperate.

Individual bacterial and yeast cells often produce molecules that are used by others. Whilst many cells share the benefits of these 'public goods', at least some cells have to endure the costs involved in producing them. As such, selfish individuals can benefit from molecules made by others, without making their own. However, if everyone cheated in this way, the public good would be lost completely: this is called the 'public goods dilemma'.

Allen et al. have developed a mathematical model of a public goods dilemma within a microbial colony, in which the public good travels from its producers to other cells by diffusion. The fate of cooperation in this 'diffusible public goods dilemma' depends on the spatial arrangement of cells, which in turn depends on their shape and the spacing between them. Other important factors include rates of diffusion and decay of the public good—both of which affect how widely the public good is shared.

The model predicts that cooperation is favored when the diffusion rate is small, when the colonies are flatter, and when the public goods decay slowly. These conditions maximize the benefit of the public goods enjoyed by the cell producing them and its close neighbors, which are also likely to be producers. Public goods dilemmas are common in nature and society, so there is much interest in identifying general principles that promote cooperation.

that spatial structure can promote cooperation by facilitating clustering and benefit-sharing among cooperators. However, these mathematical results focus largely on pairwise interactions rather than diffusible public goods. On the other hand, previous theoretical works that specifically explore microbial cooperation (*West and Buckling, 2003*; *Ross-Gillespie et al., 2007*; *Driscoll and Pepper, 2010*) use a relatedness parameter in place of an explicit spatial model, obscuring the important roles of colony geometry and spatial diffusion in determining the success of cooperation.

## Results

Here we present a simple spatial model of a diffusible public goods dilemma. Our model is inspired by the quasi-regular arrangements of cells in many microbial colonies (*Figure 1A,B*). The geometry of these arrangements depends on the shapes of cells and the dimensionality of the environment. For example, approximately spherical organisms such as *S. cerevisiae* arrange themselves in a hexagonal lattice-like structure when the colony is constrained to a two-dimensional plane (*Figure 1A*). This differs from the arrangements of rod-shaped organisms such as the bacterium *Escherichia coli* (*Figure 1B*).

To allow for a maximum variety of possible arrangements, we represent space as a weighted graph $G$ (*Figure 1C,D*; *Lieberman et al., 2005*). Edges join cells to their neighbors, with edge weights $e_{ij}$ proportional to the frequency of diffusion between neighboring cells. The graph structure thereby captures all features of cell arrangement that are relevant to the diffusion of public goods. The edge weights are normalized to satisfy $\Sigma_j e_{ij} = 1$, so that they represent relative frequencies of diffusion to each neighbor. Since we are modeling intercellular diffusion, we set $e_{ii} = 0$ for each $i$. We also suppose that $G$ has bi-transitive symmetry (*Taylor et al., 2007*), which implies that space is homogeneous (i.e., that the colony looks the same from each cell). Our model therefore applies primarily to the interiors of colonies rather than their boundaries. Bi-transitive symmetry also requires that pairwise relationships are symmetric—in particular $e_{ij} = e_{ji}$ for every pair $i$ and $j$. This captures the reasonable assumption that public goods diffuse as frequently from cell $i$ to cell $j$ as they do from $j$ to $i$.

To characterize local structure, we introduce the *Simpson degree* $\kappa = \left( \sum_{j \in G} e_{ij}^2 \right)^{-1}$. This quantity can be understood as the Simpson diversity (*Simpson, 1949*) of neighbors per cell, and coincides with the usual notion of degree on regular unweighted graphs. By symmetry, $\kappa$ does not depend on which vertex $i$ is used in the above sum.

We consider two cells types: cooperators, $C$, that produce the public good, and defectors, $D$, that do not. These traits are passed to offspring upon reproduction. Production of the public good inflicts

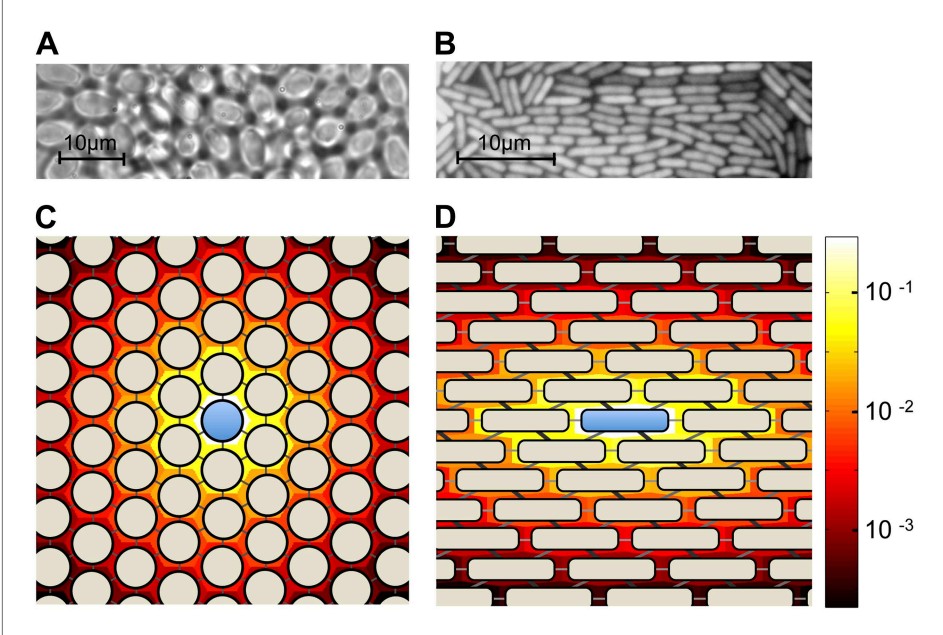

**Figure 1**. Colony geometry and public goods sharing in microbes of different shapes. (**A**) A two-dimensional colony of *S. cerevisiae* self-organizes into approximate hexagonal geometry due to the spherical shape of yeast cells. (**B**) A two-dimensional colony of *E. coli*, expressing green fluorescent protein, exhibits transient regular-graph-like structure. Panels **C** and **D** show idealized graph representations of colony spatial structure, and the consequent sharing of public goods, for sphere-shaped and rod-shaped organisms, respectively. Background colors show the stationary distributions $\psi_i$ of public goods resulting from a single cooperator (center). In each case, the diffusion parameter is set as $\lambda = 3$. (**C**) Two-dimensional colonies of spherical organisms can be represented by triangular lattices with uniform edge weights. (**D**) Two-dimensional colonies of rod-shaped organisms can be represented using a triangular lattice with unequal weights. In this case, the weights are chosen as 0.1, 0.15 and 0.25, roughly proportional to the shared surface area between *E. coli* cells when arranged as shown.

a cost $c$ on its producer, and generates a total benefit $b$ that is distributed among cells according to a diffusion process described below. Because our model is inspired by public goods that directly increase cell growth rate (such as hydrolyzed monosaccharides) it is less applicable to public goods with indirect benefits, such as quorum-sensing molecules (**Waters and Bassler, 2005**).

Cooperators produce one unit of public good per unit time. The public goods in the vicinity of a given cell either are utilized for the benefit of this cell or diffuse toward neighboring cells in proportion to edge weight. (The possibility of public goods decay is discussed below.) We quantify diffusion by the ratio $\lambda$ of the diffusion rate to the utilization rate. The dynamics of the local public goods concentration $\psi_i$ at each node $i \in G$ are given by

$$\dot{\psi}_i = s_i - \psi_i - \lambda \psi_i + \lambda \sum_{j \in G} e_{ji} \psi_j. \tag{1}$$

Above, $s_i = 0,1$ indicates the current type, $D$ or $C$ respectively, of cell $i$. The term $s_i$ in **Equation 1** represents public goods production, $-\psi_i$ represents utilization, $-\lambda \psi_i$ represents diffusion outward, and the remaining term represents diffusion inward.

**Equation 1** is equivalent to supposing that each particle of public good performs a random walk among cells (with step probabilities equal to edge weights), and has probability $1/(1+\lambda)$ of being utilized at each cell it encounters, including its producer. In this interpretation, $\lambda$ equals the expected number of steps a particle travels before being utilized.

For most empirical systems, diffusion and utilization occur much faster than cell division. We therefore suppose that the local public goods concentrations $\psi_i$ reach stationary equilibrium levels between reproductive events ('Materials and methods').

Two key quantities in our analysis are the fractions, $\phi_0$ and $\phi_1$, of public goods that are retained by its producer and the producer's immediate neighbors, respectively (**Figure 2**). For a state in which only a single cell, $i$, is a cooperator, we have $\phi_0 = \psi_i$ and $\phi_1 = \Sigma_{j \in G}\, e_{ij}\, \psi_j$.

Turning now to the dynamics of evolution, we suppose that the fecundity (reproductive rate) of cell $i$ is given by $F_i = 1 + b\psi_i - cs_i$. In words, each individual has baseline fitness 1, plus the benefit, $b\psi_i$, of public goods utilization, minus the cost, $cs_i$ of public goods production. We suppose $b > 0$ and $0 < c < 1$, so that benefits, costs, and overall fecundity are always positive. Some of our results apply to all such $b$ and $c$ values, while others apply only in the weak selection regime, $b, c \ll 1/\kappa$.

Reproductions and deaths follow the Death–Birth update rule (**Ohtsuki et al., 2006**). At each time step, a cell is selected randomly to die, with uniform probability. A neighbor of the now-vacant position is randomly selected to reproduce, with probability proportional to fecundity times edge weight. The new offspring fills the vacancy. For the moment, we suppose that reproduction follows the same edge weights as diffusion (we will relax this assumption later). We also consider other update rules in **Supplementary file 1**.

We quantify the evolutionary success of cooperation in terms of the fixation probabilities $\rho_C$ and $\rho_D$, defined as the probability that the cooperator or defector type, respectively, will fix, upon starting from a single mutant in a population initially of the opposite type. Cooperation is favored if $\rho_C > \rho_D$. This is equivalent to the condition that, for small mutation rates, cooperators have greater time-averaged frequency than would be expected from mutational equilibrium alone (**Allen and Tarnita, 2012**).

The assortment of cell types due to local reproduction can be studied using coalescing random walks (**Wakeley, 2009**; **Allen et al., 2012**), which represent the ancestral lineages of chosen individuals as the coalesce into the most recent common ancestor. By applying random walk theory to both diffusion and assortment, we are able to obtain exact conditions for the success of cooperation ('Materials and methods'; **Supplementary file 1**).

We find that public goods cooperation is favored, for any graph and diffusion rate, if and only if

$$\frac{b}{c} > \frac{1}{\phi_0 + \phi_1}. \qquad (2)$$

In words, cooperation is favored if, of the public goods a cooperator produces, the benefits received by the producer, $b\phi_0$, plus the (edge-weighted) average benefits received by each neighbor, $b\phi_1$, outweigh the cost $c$ of production (**Figure 2**). This result is strikingly simple, given the complex patterns of public goods sharing that result from diffusion (**Figure 1**). Condition (**2**) holds for arbitrary selection strength on complete graphs and one-dimensional lattices, and for weak selection on other graphs. This condition also holds for a variety of other diffusion processes (**Supplementary file 1**)—including diffusion that follows a different graph structure from reproduction. (In this case, the neighbor average $\phi_1$ is computed using the weights for the reproduction graph.)

Condition (**2**) can alternatively be expressed as $b/c > \lambda/[\phi_0\,(1 + 2\lambda) - 1]$ ('Materials and methods'), showing how the success of cooperation depends on the relationship between the retention fraction $\phi_0$ and the diffusion parameter $\lambda$. We have derived this relationship exactly for simple graph structures (**Table 1**), and present a general method for obtaining this relationship in the 'Materials and methods'. **Figure 3A,B** illustrates how the critical $b/c$ ratios vary with the diffusion parameter $\lambda$ and the graph topology.

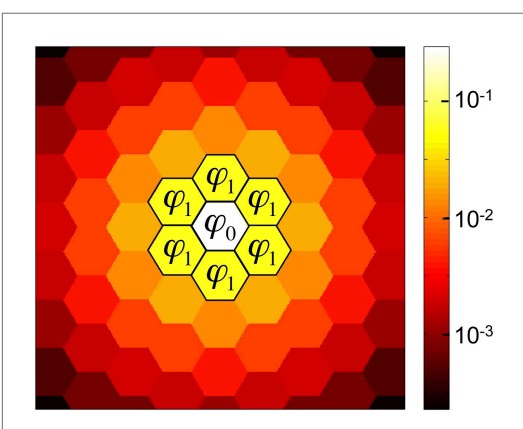

**Figure 2**. The success of cooperation depends on the amounts of public good retained by a cooperator and its neighbors. Of the public good that a cooperator produces, a fraction $\phi_0$ is retained by the producer, a fraction $\phi_1$ is absorbed by each of the cooperator's nearest neighbors, and the remainder diffuses to cells further away. (For graphs with unequal edge weights, $\phi_1$ is the edge-weighted average fraction received by each neighbor.) Cooperation is favored if $b/c > 1/(\phi_0 + \phi_1)$, that is, if the benefit $b\phi_0$ received by producer, plus the average benefit $b\phi_1$ received by each neighbor, exceeds the cost $c$ of production. This figure shows a triangular lattice with equal edge weights and diffusion parameter $\lambda = 3$.

**Table 1.** Fraction of public goods retained by producer for different graph structures and diffusion rates

| Graph structure* | Fraction $\phi_0$ of public goods retained |
|---|---|
| Complete (well-mixed) | $\dfrac{1}{1+\lambda}$ |
| 1D lattice | $\dfrac{1}{\sqrt{1+2\lambda}}$ |
| 2D square lattice† | $\dfrac{1}{\mathrm{agm}(1,1+2\lambda)}$ |
| $n$-dimensional lattice (general)‡ | $\dfrac{1}{(2\pi)^n}\int_{-\pi}^{\pi}\cdots\int_{-\pi}^{\pi}\dfrac{d^n\mathbf{y}}{1+\lambda-\lambda\,\chi(\mathbf{y})}$ |
| $k$-Bethe lattice§ | $\dfrac{\sqrt{(k-2)^2(1+\lambda)^2+4(k-1)(1+2\lambda)}-(k-2)(1+\lambda)}{2(1+2\lambda)}$ |

*These results are for large populations. Corrections for finite population size are given in **Supplementary file 1**.
†agm denotes the arithmetic-geometric mean.
‡This result applies to any mathematical lattice, including triangular and von Neumann lattices. $\chi(\mathbf{y})$ denotes the structure function of the lattice in question, defined in the 'Materials and methods'.
§A Bethe lattice (a.k.a. infinite Cayley tree), is an infinite regular graph with no cycles. In the formula, $k$ denotes the graph degree.

Above, we have assumed that diffusion and replacement are both described by the same graph structure. However, this may not be the case for all microbes. In *E. coli* colonies, for example, it is reasonable to conjecture that diffusion occurs more frequently among cells that have a long side in common, whereas replacement may occur more frequently among end-to-end neighbors (**Figure 1A,C**). Additionally, some systems may follow a public goods diffusion process other than that modeled by **Equation 1**. To account for these variations, we consider a more general model in which diffusion is described by the fractions $\phi_{ij}$ of public goods which, if produced by cell $i$, would be utilized by cell $j$. Probabilities of replacement are described by a graph with edge weights $e_{ij}$ as before. The diffusion fractions $\phi_{ij}$ are normalized so that $\sum_j \phi_{ij} = 1$ for each $i$, and they have the same symmetries as the replacement graph; within these restrictions, they may be specified arbitrarily. Remarkably, our main result, **Equation 1**, remains valid in this generalized setting, with the neighbor average $\phi_1$ defined as $\phi_1 = \sum_j e_{ij}\,\phi_{ij}$.

## Discussion

Our results suggest three qualitative regimes for diffusible public goods scenarios. For $\lambda \ll 1$, the benefits are almost all retained by producer, and production is favored whenever $b/c > 1$. Conversely, for $\lambda \ll 1$, public goods are shared indiscriminately, and production is favored only if public goods are essential for survival, in which case $b$ is effectively infinite. Between these extremes, public goods are shared locally, and the spatial arrangement of cells plays a critical role in the success of cooperation (**Figure 3A**). At the smaller end of this critical regime, the expansion $b/c > 1+\lambda(\kappa-1)/\kappa+\mathcal{O}(\lambda^2)$ of condition (**2**), derived in **Supplementary file 1**, shows how the difficulty of cooperation increases with the diffusion parameter $\lambda$ and the Simpson degree $\kappa$. For the hydrolysis of monosaccharides in *S. cerevisiae*, we estimate $\lambda \sim 3$ ('Materials and methods'); thus we expect the success of invertase production to be strongly affected by colony geometry. Interestingly, this diffusion length is of the same order of magnitude as those reported in other recent experiments with diffusible public goods (**Julou et al., 2013**; **Momeni et al., 2013**).

Our model predicts that the advantage of cooperation decreases with colony dimensionality; for example, less cooperation would be expected in three-dimensional structures than in flat (2D) colonies (**Figure 3A**). It also predicts that cooperation becomes more successful with increased viscosity of the environment and/or rate of public goods utilization, both of which would decrease $\lambda$.

A more subtle question is how cooperation is affected if the public good may decay (or equivalently, escape the colony) instead of being utilized. Decay reduces the absolute amount of public

goods to be shared, but also restricts this sharing to a smaller circle of neighbors; thus the net effect on cooperation is at first glance ambiguous. We show in the 'Materials and methods' that incorporating decay effectively decreases $\lambda$ by a factor $1/(1 + d)$, reflecting the smaller neighborhood of sharing, and also effectively decreases $b$ by the same factor, reflecting the diminished absolute amount of public goods. Here $d$ represents the ratio of the decay rate to the utilization rate. Since the critical benefit-to-cost ratio always increases sublinearly with $\lambda$, the net effect is to make cooperation more difficult (see *Figure 3C*). Thus decay of the public good has a purely negative effect on cooperation.

Our results help elucidate recent emiprical results on microbial cooperation in viscous environments. For example, *Kümmerli et al. (2009)* found that increased viscosity promotes the evolution of siderophore production in *Pseudomonas aeruginosa*, while *Le Gac and Doebeli (2010)* found that viscosity had no effect on the evolution of colicin production in *E. coli*. In both cases, increased viscosity restricted cell movement, effectively leading to fewer neighbors per cell (lower graph degree). The crucial difference lies in the effect on public goods diffusion. In the study of *Kümmerli et al. (2009)*, the diffusion rate decreased significantly as viscosity increased, while for *Le Gac and Doebeli (2010)*, the diffusion rate remained large even with high viscosity. Thus the divergent outcomes can be understood as a consequence of differences in the diffusion rate, captured in our model by $\lambda$.

Here we have considered homotypic cooperation—cooperation within a single population. *Momeni et al. (2013)*, published previously in *eLife*, investigate heterotypic cooperation between distinct populations of *S. cerevisiae*, in the form of exchange of essential metabolites. Type *R* produces adenine and requires lysine, type *G* produces lysine and requires adenine, and type *C* (a cheater) requires adenine but does not produce adenine. While such heterotypic cooperation is not incorporated in our model, the results are qualitatively similar, in that spatial structure promoted the cooperative strategies *G* and *R* over the cheater *C*. This similarity can be understood by noting that heterotypic cooperation also entails a form of second-order homotypic cooperation. For example, *G*-cells aid nearby *R*-cells, which in turn aid nearby *G*-cells, so the benefit produced by a *G*-cell indirectly aids other *G*-cells nearby. Thus the conclusion that spatial structure aids cooperative strategies can apply to heterotypic cooperation as well.

Finally, our model can also represent the spread of behaviors via imitation on social networks (*Bala and Goyal, 1998*; *Bramoullé and Kranton, 2007*; *Christakis and Fowler, 2007*). Suppose an action generates a benefit $b_0$ for the actor, and additionally generates further benefits that radiate outward according to some multiplier $m$, so that first neighbors receive a combined benefit $mb_0$, second neighbors receive $m^2 b_0$, and so on. Education, for example, exhibits this kind of social multiplier in its effect on wages (*Glaeser et al., 2003*). This effect can be captured using the parameter change $b = b_0/(1 - m)$, $\lambda = m/(1 - m)$. For non-well-mixed social networks, the action becomes more likely to spread as the multiplier increases, and can spread even if there is a net cost to the actor (*Figure 4*).

## Materials and methods

### Stationary public goods distribution

We obtain a recurrence relation for the stationary public goods distribution in a given state by setting $\dot{\psi}_i = 0$ in *Equation 1*. This yields

$$(1+\lambda)\psi_i = s_i + \lambda\sum_{j\in G}e_{ji}\psi_j. \tag{3}$$

In particular, for a state in which only cell $i$ is a cooperator, we have $(1 + \lambda)\phi_0 = 1 + \lambda\phi_1$. Combining this identity with (*2*) yields the equivalent condition $b/c > \lambda[\phi_0 (1 + 2\lambda) - 1]$.

### Generating function analysis of random walks

We analyze the distribution of public goods and the assortment of cell types using the generating function for random walks (*Montroll and Weiss, 1965*; *Lawler and Limic, 2010*). For a given graph $G$, this generating function is given by the power series

$$\mathcal{G}_{ij}(z) = \sum_{n=0}^{\infty} p_{ij}^{(n)} z^n.$$

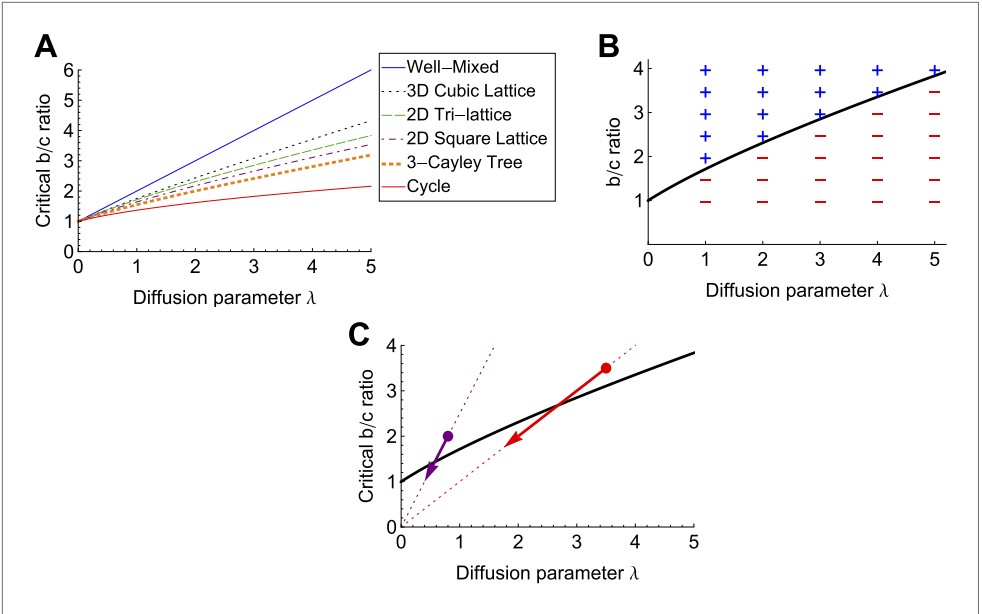

**Figure 3**. Cooperation becomes harder to achieve with increasing $\lambda$, graph degree and dimensionality, and public goods decay rate. (**A**) The critical $b/c$ ratio for public goods production to be favored for various graph structures, plotted against the diffusion rate $\lambda$. These results are derived from **Equation 2** and the expressions for $\phi_0$ in **Table 1**. For a well-mixed population (complete graph), cooperation is favored if and only if $b/c > 1 + \lambda$; for other graph structures, the critical $b/c$ ratio is a increasing, convex function of $\lambda$. In general, the conditions for cooperation become increasingly stringent with both the degree and the dimensionality of the graph. (**B**) Our results are confirmed by simulations on a 15 × 15 periodic triangular lattice with uniform edge weights and cost $c$ = 5%. The critical $b/c$ threshold from **Equation 2** is plotted in black. A plus (+) indicates that the frequency of cooperator fixation exceeded the frequency of defector fixation ($\rho_C > \rho_D$), while a minus (−) indicates the opposite. In all cases the results were statistically significant (two-proportion pooled $z$-test, $p<0.05$). (**C**) Adding decay of rate $d$ effectively reduces both $\lambda$ and $b$ by the factor $1/(1 + d)$, reflecting greater locality in sharing but reduced overall concentration of public good. On a graph of $b/c$ versus $\lambda$, this moves each point ($b/c$, $\lambda$) along a straight line toward the origin. Since the increase in the critical $b/c$ ratio with $\lambda$ is in all cases sublinear, this change always hinders cooperation. The critical $b/c$ ratio for a planar triangular lattice is plotted in black. Adding a decay rate equal to the utilization rate ($d$ = 1) changes favorable ($b/c$, $\lambda$) combinations (marked by circles) to unfavorable ones (arrowheads).

Above, $p_{ij}^{(n)}$ denotes the probability that a random walk of $n$ steps starting at $i$ will terminate at $j$. We prove in **Supplementary file 1** that the stationary concentration of public goods in a particular state are given by

$$\psi_i = \frac{1}{1+\lambda}\sum_{j\in G}s_j\,\mathcal{G}_{ji}\left(\frac{\lambda}{1+\lambda}\right).$$

In particular, the fraction $\phi_0$ that a cooperator retains of its own public good can be written

$$\phi_0 = \frac{1}{1+\lambda}\mathcal{G}_{ii}\left(\frac{\lambda}{1+\lambda}\right). \tag{4}$$

Spatial assortment of types can be quantified using identity-by-descent IBD probabilities (**Rousset and Billiard, 2000**; **Taylor et al., 2007**). For this, we introduce a small probability $u$ that each new offspring is a mutant. Then, two given cells are IBD if no mutation separates them from their most recent common ancestor. Based on the theory of coalescing random walks (**Allen et al., 2012**), the probability that $i$ and $j$ are IBD can be written

$$q_{ij} = \frac{\mathcal{G}_{ij}(1-u)}{\mathcal{G}_{jj}(1-u)}.$$

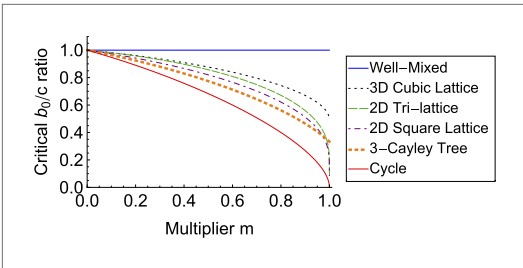

**Figure 4**. The spread of behaviors on social networks increases with their social multipliers. In an alternate interpretation of our model, an action has benefits that radiate outward from the actor according to some multiplier *m*. Individual receiving a large amount of benefit are more likely to be imitated by social contacts. The likelihood of the action to spread—and the benefits to the network as a whole—both increase with the multiplier *m*.

Considering the dynamics of Death–Birth updating, and applying established properties of generating functions, we derive (**Supplementary file 1**) the success condition (2).

To obtain the expressions in **Table 1**, we combine (4) with previously established expressions for $\mathcal{G}_{ij}$ on the graphs in question. A general expression is available for a lattice of any dimension. Such a lattice is defined by a finite collection of vectors $v_1,\ldots,v_k \in \mathbf{R}^n$ with associated weights $w_1,\ldots,w_k$. The nodes of the lattice are all points of the form $\mathbf{x} = m_1 v_1 + \ldots + m_k v_k \in \mathbf{R}^n$, where $m_1,\ldots,m_k$ are integers. The edges from a node $\mathbf{x}$ consist of the vectors $v_1, \ldots, v_k$, positioned to start at the point $\mathbf{x}$, with weights given by $w_1, \ldots, w_k$, respectively. The generating function of a random walk on such a lattice, starting from the lattice origin **0**, can be expressed as (**Montroll and Weiss, 1965**)

$$\mathcal{G}_{0\mathbf{x}}(z) = \frac{1}{(2\pi)^n} \int_{-\pi}^{\pi} \cdots \int_{-\pi}^{\pi} \frac{e^{-i\mathbf{x}\cdot\mathbf{y}}}{1 - z\,\chi(\mathbf{y})} d^n\mathbf{y}. \qquad (5)$$

Above, $\chi(\mathbf{y})$ is the 'structure function' of the lattice, defined as

$$\chi(\mathbf{y}) = \sum_{j=1}^{k} w_k e^{i\mathbf{v}_j \cdot \mathbf{y}}. \qquad (6)$$

The argument $\mathbf{y} = (y_1,\ldots,y_n)$ of $\chi(\mathbf{y})$ is a vector in $\mathbf{R}^n$. For example, for an *n*-dimensional square lattice, we have

$$\chi(\mathbf{y}) = \frac{1}{n} \sum_{i=1}^{n} \cos(y_i).$$

For a two-dimensional triangular lattice,

$$\chi(\mathbf{y}) = \frac{1}{3}\Big[\cos(y_1) + \cos(y_2) + \cos(y_1 + y_2)\Big].$$

Similar expressions for other lattices, including the square lattice with von Neumann neighbors and lattices with unequal edge weights (e.g., **Figure 1B**), can be readily obtained from (6).

## Estimation of diffusion parameter for *S. cerevisiae*

We suppose that glucose uptake follows Michaelis–Menten kinetics, so that the uptake rate is given by $V_{max}\psi/(\psi+K)$, where $\psi$ is the concentration of glucose, $V_{max}$ is the maximal uptake rate, and $K$ is the concentration at which the uptake rate reaches half of its maximum. We treat fructose as equivalent to glucose. Since we are interested in the case that glucose is limited, we assume $\psi \ll K$, and the uptake rate therefore simplifies to $V_{max}\psi/K$. **Gore et al. (2009)** estimated the uptake kinetics to be $V_{max} \sim 2 \times 10^7$ molecules per second and $K \sim 1mM$.

We calculate the lifetime *L* of a glucose molecule prior to absorption as the reciprocal of the fraction of glucose absorbed per unit time:

$$L = \frac{\#\,\text{glucose molecules per unit excluded volume}}{(\text{Uptake rate per cell}) \times (\#\,\text{cells per unit excluded volume})},$$

where 'excluded volume' refers to the volume of water excluded by the yeast cells. Supposing that each yeast cell has volume $v \sim 4\pi(2\mu m)^3/3$, and that yeast cells in a tightly-packed colony occupy approximately half of the available volume, we obtain

$$L = \frac{\psi}{(V_{max}\,\psi/K)\times(1/v)}$$

$$= \frac{Kv}{V_{max}}$$

$$\sim 1\,\text{sec.}$$

The diffusion length before uptake is calculated as $\sqrt{D/L}$, where $D$ is the diffusion constant, which we estimate as 100 $\mu m^2$/sec in the colony environment. Combining with the above calculation of $L$ gives a diffusion length of ~10 $\mu m$, which is ~3 cell lengths. We therefore estimate $\lambda = 3$ for this system.

### Decay of the public good

Decay or escape of the public good can be incorporated into our model by adding a decay term to the right-hand side of *Equation 1*. This yields

$$\dot{\psi}_i = s_i - \psi_i - d\psi_i - \lambda\psi_i + \lambda\sum_{j\in G} e_{ji}\psi_j.$$

Above, $d$ represents the ratio of the decay rate to the utilization rate. Setting $\dot{\psi}_i = 0$ and rearranging, we obtain

$$\psi_i(1+d)\left(1+\frac{\lambda}{1+d}\right) = s_i + \frac{\lambda}{1+d}\sum_{j\in G} e_{ji}\psi_j(1+d).$$

Defining the effective quantities $\tilde{\psi}_i = \psi_i(1+d)$ and $\tilde{\lambda} = \lambda/(1+d)$, we recover the recurrence relation (*3*). All of our results then carry forward using these effective quantities, except that $b$ must also be reduced by the factor $1 + d$ to compensate for the rescaling of $\psi_i$ by this same factor.

## Acknowledgements

We thank Andrea Velenich for obtaining images of *E. coli* and *S. cerevisiae* colonies. This work was supported by an NIH R00 Pathways to Independence Award (NIH R00 GM085279-02), an NIH New Innovator Award (NIH DP2), an NSF CAREER Award, a Sloan Research Fellowship, the Pew Scholars Program and the Allen Investigator Program. The Foundational Questions in Evolutionary Biology initiative at Harvard University is supported by a grant from the John Templeton Foundation.

## Additional information

### Funding

| Funder | Grant reference number | Author |
|---|---|---|
| National Institutes of Health | NIH R00 GM085279-02 | Jeff Gore |
| National Science Foundation | | Jeff Gore |
| Alfred P Sloan Foundation | | Jeff Gore |
| Pew Scholars Program | | Jeff Gore |
| Allen Investigator Program | | Jeff Gore |
| John Templeton Foundation– Foundational Questions in Evolutionary Biology | RFP-12-02 | Benjamin Allen, Martin A Nowak |
| National Institutes of Health | NIH DP2 | Jeff Gore |

The funders had no role in study design, data collection and interpretation, or the decision to submit the work for publication.

### Author contributions

BA, JG, MAN, Conception and design, Acquisition of data, Analysis and interpretation of data, Drafting or revising the article

## Additional files

**Supplementary files**

• Supplementary file 1. This file contains mathematical proofs of several results.

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
