## [Decision Letter]

Thank you for sending your work entitled “Spatial dilemmas of diffusible public goods” for consideration at *eLife*. Your article has been favorably evaluated by a Senior editor and 3 reviewers, one of whom, Carl Bergstrom, is a member of our Board of Reviewing Editors.

The editors and the reviewers discussed their comments before we reached this decision, and the Senior editor has assembled the following comments to help you prepare a revised submission.

Microbes frequently face public goods dilemmas and often these dilemmas involve the production of diffusible products secreted into the extracellular environment. It is an interesting and open question to determine when and how such behavior will be favored by natural selection. The manuscript provides a technically sound and elegant mathematical analysis of the problem based on an implicit graphical structure in the spatial organization of cells that make up a colony. The main mathematical result, which is inequality (2), is nice in its simplicity and how it incorporates the three different factors within a single representation.

The reviewers had two major concerns that need to be addressed before the manuscript can be accepted:

A) Whether the paper is of sufficient biological interest to merit publication in *eLife*. To this concern the following four comments/suggestions were provided by the reviewers:

1) While graphs provide a nice means of modeling some types of structure, one reviewer was less convinced that they are a natural way to model the structure of diffusing public goods. This approach that the authors have developed extensively over the years appears forced upon the biology of the problem rather than being an natural way to model natural interactions.

2) Do the results tell us much beyond what we already know in terms of the biological problem? For example, similar effects of the diffusion rate are already known from other models of public goods (some of which are cited), and the colony dimension results (which sounds really interesting at first) is also pretty obvious once it becomes clear what is meant by colony dimension. The main new insight about biology provided by the results is the role of the decay rate of the public good. To my knowledge at least, this idea has not previously been explored and it is clear that the tension between the various ways that decay rate enters the problem requires the sort of quantitative analysis presented here. Regardless, the result does seem like a rather modest advance in our understanding of the evolutionary interplay between public goods, diffusion, cooperation, etc.

3) The authors might also want to delve more deeply into the literature on public goods to better position their results within the existing literature. For example, there is good work by Brown, Taylor, Buckling, West, and others. Some of this is cited but not discussed in a very thorough way, and some is not even cited. A none-exhaustive list of other potentially useful papers include:

Buckling, A, Harrison, F, Vos, M, Brockhurst, MA, Gardner, A, West, SA & Griffin, AS. 2007 Siderophore-mediated cooperation and virulence in Pseudomonas aeruginosa. FEMS Microbiolol. Ecol. 62, 135-141. doi:10.1111/j.1574-6941.2007.00388.x

West, SA & Buckling, A. 2003 Cooperation, virulence and siderophore production in bacterial parasites. Proc. R. Soc. Lond. B 270, 37-44. doi:10.1098/rspb.2002.2209

Bramoulle, Y & Kranton, R. Public goods in networks. Journal of Economic Theory. 135 (1), 478-494

4) You may also be able to address this concern by referencing and coordinating the text of your paper with the parallel submission by Shou et al.

B) Avoid confusion about the use of the term “Bethe Lattice”. A reviewer provided the following commentary/suggestions:

“In order to avoid later confusion I suggest substituting the expression “Bethe lattice” or “locally Cayley tree structure” for “Cayley tree” through the whole text. For finite Cayley trees a relevant portion of the nodes are located on the periphery where each node has only one neighbor. This is the reason why the behavior of the Ising model on the Cayley tree is similar to those observed on the one-dimensional chain (no magnetic ordering at finite temperatures). On the contrary, the Ising model on Bethe lattice exhibits a mean-field type order-disorder phase transition (when increasing the temperature) that can be described exactly by several methods, e.g., by the cavity method or pair approximation [for details see the review by Dorogovtsev et al., Rev. Mod. Phys. 80 (2008) 1275-1335]. The concept of Bethe lattice neglects the effects of periphery and involves equivalence between the nodes, as it is assumed in the present work, too.”

---

## [Author Response]

*The reviewers had two major concerns that need to be addressed before the manuscript can be accepted*:

*A) Whether the paper is of sufficient biological interest to merit publication in eLife. To this concern the following four comments/suggestions were provided by the reviewers*:

*1) While graphs provide a nice means of modeling some types of structure, one reviewer was less convinced that they are a natural way to model the structure of diffusing public goods. This approach that the authors have developed extensively over the years appears forced upon the biology of the problem rather than being an natural way to model natural interactions*.

While they may have an abstract “flavor”, graphs are a very natural tool for representing a wide variety of spatial relationships. Compared, for example, to lattice models (an accepted tool of the field), graphs have more flexibility to represent the distinct patterns of cell arrangement that occur in microbial colonies. In this study we use weighted graphs to allow for different diffusion rates between different kinds of neighbors (e.g., lateral versus end-to-end). The symmetry assumptions correspond to the quasi-regular structures that are often found in colony interiors.

*2) Do the results tell us much beyond what we already know in terms of the biological problem? For example, similar effects of the diffusion rate are already known from other models of public goods (some of which are cited), and the colony dimension results (which sounds really interesting at first) is also pretty obvious once it becomes clear what is meant by colony dimension. The main new insight about biology provided by the results is the role of the decay rate of the public good. To my knowledge at least, this idea has not previously been explored and it is clear that the tension between the various ways that decay rate enters the problem requires the sort of quantitative analysis presented here. Regardless, the result does seem like a rather modest advance in our understanding of the evolutionary interplay between public goods, diffusion, cooperation, etc*.

In addition to our results on the effects of the decay rate, our model makes the unexpected prediction that the success of cooperation depends only on the amounts of public goods received by a cell and its immediate neighbors. Thus, even though public goods may be shared at arbitrarily large distances, the success of this behavior can be understood by examining neighbors at distance one.

*3) The authors might also want to delve more deeply into the literature on public goods to better position their results within the existing literature. For example, there is good work by Brown, Taylor, Buckling, West, and others. Some of this is cited but not discussed in a very thorough way, and some is not even cited*.

We thank the reviewers for the suggestions. We have incorporated the suggested references, along with others that have appeared recently. We now discuss these contributions in greater detail in the last paragraph of the Introduction. We have also incorporated a recent study of diffusible public goods by [18] into our references and Discussion.

*4) You may also be able to address this concern by referencing and coordinating the text of your paper with the parallel submission by Shou et al*.

We have added an exploration of the parallels of our work with that of [26] at the second-to-last paragraph of the Discussion. Although the work of [26] concerns heterotypic cooperation—which is not directly represented in our model—we present a new argument that heterotypic cooperation in space also entails a kind of second-order homotypic cooperation, so that results from models like ours can also shed light on heterotypic cooperation, as investigated by [26].

*B) Avoid confusion about the use of the term “Bethe Lattice”. A reviewer provided the following commentary/suggestions*:

*“In order to avoid later confusion I suggest substituting the expression “Bethe lattice” or “locally Cayley tree structure” for “Cayley tree” through the whole text. For finite Cayley trees a relevant portion of the nodes are located on the periphery where each node has only one neighbor. This is the reason why the behavior of the Ising model on the Cayley tree is similar to those observed on the one-dimensional chain (no magnetic ordering at finite temperatures). On the contrary, the Ising model on Bethe lattice exhibits a mean-field type order-disorder phase transition (when increasing the temperature) that can be described exactly by several methods, e.g., by the cavity method or pair approximation [for details see the review by Dorogovtsev et al., Rev. Mod. Phys. 80 (2008) 1275-1335]. The concept of Bethe lattice neglects the effects of periphery and involves equivalence between the nodes, as it is assumed in the present work, too.*”

We apologize for this confusion. We now use the term Bethe lattice throughout.